# Analyticity of critical exponents of the $O(N)$ models from nonperturbative renormalization

**Andrzej Z. Chlebicki and Pawel M. Jakubczyk⋆**

Institute of Theoretical Physics, Faculty of Physics, University of Warsaw,
Pasteura 5, 02-093 Warsaw, Poland

⋆ pawel.jakubczyk@fuw.edu.pl

## Abstract

We employ the functional renormalization group framework at the second order in the derivative expansion to study the $O(N)$ models continuously varying the number of field components $N$ and the spatial dimensionality $d$. We in particular address the Cardy-Hamber prediction concerning nonanalytical behavior of the critical exponents $\nu$ and $\eta$ across a line in the $(d, N)$ plane, which passes through the point $(2, 2)$. By direct numerical evaluation of $\eta(d, N)$ and $\nu^{-1}(d, N)$ as well as analysis of the functional fixed-point profiles, we find clear indications of this line in the form of a crossover between two regimes in the $(d, N)$ plane, however no evidence of discontinuous or singular first and second derivatives of these functions for $d > 2$. The computed derivatives of $\eta(d, N)$ and $\nu^{-1}(d, N)$ become increasingly large for $d \to 2$ and $N \to 2$ and it is only in this limit that $\eta(d, N)$ and $\nu^{-1}(d, N)$ as obtained by us are evidently nonanalytical. By scanning the dependence of the subleading eigenvalue of the RG transformation on $N$ for $d > 2$ we find no indication of its vanishing as anticipated by the Cardy-Hamber scenario. For dimensionality $d$ approaching 3 there are no signatures of the Cardy-Hamber line even as a crossover and its existence in the form of a nonanalyticity of the anticipated form is excluded.



# 1   Introduction

The $O(N)$ models count among the most paradigmatic systems in the theory of critical phenomena and were with great success applied to address universal characteristics of an amazingly broad variety of physical situations [1,2]. Even though the physically most relevant cases correspond to integer number of order-parameter components $N$ and integer spatial dimensionality $d$, it has proven extremely fruitful to consider these quantities formally as continuous parameters, leading to the development of theoretical approaches such as the $(4-\epsilon)$-expansion, $(2+\epsilon)$-expansion, or the $1/N$-expansion, where one accesses the most relevant range of parameters ($d = 3$ in particular) by expanding around an analytically soluble point in the $(d, N)$-plane. It is also worthwhile observing that there has recently been certain interest (both experimental and theoretical) in engineering situations, where the effective dimensionality of the system would not coincide with the physical dimensionality and, in particular, might take a fractional value (see e.g. Refs. [3–5]). Also note that mathematically rigorous meaning can be provided for continuous range of $N$ [6].

A very peculiar physical situation corresponds to $(d, N) = (2, 2)$, representing the Kosterlitz-Thouless (KT) universality class [7,8]. The vicinity of this point in the $(d, N)$ plane is schematically illustrated in Fig. 1. By infinitesimal variations of $(d, N)$ from the KT point one changes drastically the system behavior, and the anticipated character of this change heavily depends on the direction. The KT universality class is itself very special due to its unique, vortex unbinding driven mechanism of the phase transition. The behavior of the correlation length is controlled by an essential singularity rather than a power law, making it distinct from the transition at any $d > 2$. It follows that the critical exponent $\nu(d, N = 2)$ diverges for $d \to 2^+$.

The KT case $(d, N) = (2, 2)$ is analytically tractable and it is natural to adopt the $d = 2 + \epsilon$ expansion in an attempt to access also higher dimensionalities. It was this approach that was pursued [10] by Cardy and Hamber and led to the prediction of the existence of a line [hereafter referred to as Cardy-Hamber (C-H) line] in the $(d, N)$ plane across which the critical exponents would not be analytical functions of $(d, N)$. The procedure adopted in Ref. [10] combines the equations studied before by Nelson and Fisher [11] (valid for $N = 2$, $d \geq 2$ and constituting an extension of the KT equations) with those analyzed by Brézin and Zinn-Justin [12] (valid for $N > 2$ and zero vortex fugacity $y^2$). Under the assumption of analyticity, one may simply add up the beta functions of the renormalization group (RG) equations from both these studies and interpolate between the two limiting cases. In Ref. [10], this reasoning led to a set of equations for $y^2$ and the interaction coupling $g$ expanded to the order $\mathcal{O}(d - 2, N - 2, y^2)$.

The predicted nonanalyticity of the critical exponents arises due to the existence of two distinct solutions to the fixed-point equations. Each of the solutions is physical and describes a critical point only in a restricted region of the $(d, N)$-plane. The boundary between these regions defines the C-H line. At the approximation level of Ref. [10], across the C-H line,

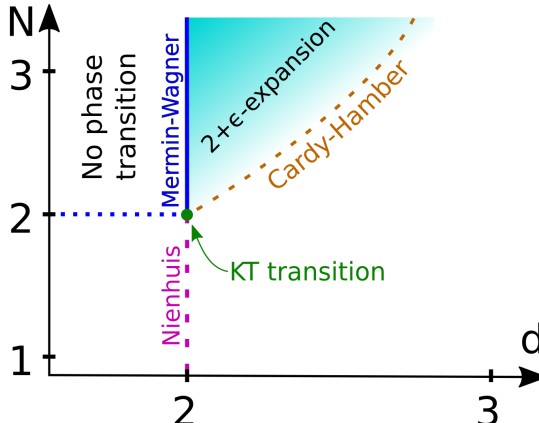

Figure 1: (Color online) The $(d,N)$ plane in the vicinity of the Kosterlitz-Thouless (KT) point $(2,2)$, schematically illustrating the landscape of universality classes with some of its most characteristic features. The Mermin-Wagner line separates the regions with possible and impossible symmetry-breaking phase transitions for $N > 2$. The critical exponents are expected to be non-analytical across the Cardy-Hamber line, which separates the regimes characterized by irrelevant ($N$ large) and relevant ($N$ small) vortices. Exact expressions for the $\nu^{-1}$ and $\eta$ exponents are available along the Nienhuis line $[d = 2, N \in (-2,2)]$ as well as in the limit $N \to \infty$ for $d > 2$. The region corresponding to $d < 2$ and $N < 2$ [9] is not discussed in the present paper.

the fixed points collide, which leads to the nonanalyticity of the critical exponents. One consequence [10] of the supposed nonanalyticity is the restriction of applicability of the $2 + \epsilon$-expansion to the region above the C-H line (see Fig. 1). As a result of truncating at leading order in $\epsilon = d - 2$, the Cardy-Hamber study does not fully characterize this predicted nonanalyticity. The shape of the C-H line is also evaluated only in a linear approximation around $(d,N) = (2,2)$; it is nonetheless expected to survive also for higher $N$, crossing $N = 3$ somewhat below $d = 3$, and even extending towards $N \to \infty$. The reason for its absence in $1/N$ calculations was attributed [10] to the non-perturbative nature of this aspect at $N$ large. To our knowledge, the C-H prediction was thus far not addressed within any alternative theoretical framework. We are also not aware of a systematic derivation of the analyzed flow equations, in particular of any studies going beyond the leading order in the $2+\epsilon$ expansion implemented by Cardy and Hamber.

In the present paper we revisit the issue of analyticity of the critical exponents from the point of view of nonperturbative RG applied to the $\phi^4$ theory. Our motivation follows primarily from the fact that (to the best of our knowledge) the shape of the C-H line seems to have never been calculated beyond the linear order in $\epsilon$. Neither was the character of the expected nonanalyticity of the critical exponents quantified. With this in mind, employing the nonperturbative RG and the derivative expansion (DE) at order $\partial^2$, we have scanned the dependence of the critical exponents $\eta$ and $\nu^{-1}$ on $(d,N)$, with particular focus on the limit $(d,N) \to (2,2)$, taken along different paths. Our results clearly indicate two distinct regimes in the $(d,N)$ plane predicted by the C-H calculation, but no evidently nonanalytical behavior (that would be visible as singularities or discontinuities of any of the first two derivatives) except for $(d,N) = (2,2)$. The computed derivatives of $\nu^{-1}(d,N)$ and $\eta(d,N)$ exhibit maxima of magnitude divergent for $(d,N) \to (2,2)$ along a line in the $(d,N)$ plane. This locus of maxima turns out to be situated not far from the expected position of the C-H line for $(d,N) \approx (2,2)$, however rapidly smoothens and vanishes completely upon increasing dimensionality towards

$d = 3$, where our calculation becomes progressively more accurate. Another key signature anticipated at the C-H line is the vanishing of the subdominant eigenvalue $e_2$ of the linearized RG transformation marking the collision with another (multicritical) fixed point. Our calculation allows for a reliable estimate of $e_2$ for $d$ separated from 2 ($d \gtrsim 2.2$) and yields no signatures of an approach of $e_2$ towards zero.

In addition to evaluating the exponents, we inspect the structure of the fixed-points located in the functional space (depending on $d$ and $N$). We recover a rapid change of the fixed-point profiles upon crossing the C-H line, which reflects the onset of vortex-dominated behavior. This is particularly transparent in the longitudinal stiffness coefficient, which exhibits a violent increase above the C-H line. There is however no signature of nonanalyticity of the fixed-point profiles marking a fixed point collision.

The paper is structured as follows: In Sec. 2 we briefly review the Cardy-Hamber approach leading to the predicted nonanalyticity of critical exponents $\eta(d, N)$ and $v^{-1}(d, N)$. In Sec. 3 we discuss the (subsequently applied) truncation of functional RG relying on the derivative expansion. In Sec. 4 we restrict to dimensionality $d = 2$, where the functional forms of the exponents $v^{-1}(d = 2, N)$ and $\eta(d = 2, N)$ are exactly known for $N < 2$. We compare our results obtained at order $\partial^2$ of the DE to the exact values. In Sec. 5 we analyze the numerically extracted profiles of the critical exponents and provide a connection to the C-H prediction. We in particular demonstrate the smoothening of $v^{-1}(d, N)$ and $\eta(d, N)$ upon moving away from $(d, N) = (2, 2)$ and emphasize that [after excluding the immediate vicinity of $(d, N) = (2, 2)$] the first two derivatives of these functions show no clear signatures of singular behavior. We identify nonetheless two regimes of the $(d, N)$ plane characterized by distinct (large-$N$-like and small-$N$-like) behavior of the critical exponents, in full consistency with the known results. The crossover between these two is very sharp for $(d, N) \approx (2, 2)$, but rapidly smoothens upon increasing the dimensionality $d$. For $d > 2.2$ we additionally present the evolution of the subdominant eigenvalue $e_2$ interpolating between $N = 2$ and $N \to \infty$. Contrary to the C-H prediction $e_2$ remains well separated from zero for all $N$. In Sec. 6 we analyze the obtained functional fixed-point profiles, demonstrating the rapid (however smooth) change across the C-H line with no indication of a collision with a different fixed-point. Sec. 7 contains summary and conclusion.

## 2 The Cardy-Hamber approach

The RG equations analyzed in Ref. [10] are given as

$$\begin{cases} \dot{g} = -\epsilon g + (N-2)f(g) + 4\pi^3 y^2 + \dots \\ \dot{y^2} = \left(4 - \frac{2\pi}{g}\right) y^2 + \dots \end{cases} \tag{1}$$

and combine the equations studied by Nelson and Fisher [11] [obtained by putting $N = 2$ in Eq. (1)] with those considered by Brézin and Zinn-Justin [12] [recovered for zero $y^2$ from Eq. (1)]. Here $g$ is the interaction coupling, and $f(g) = \frac{g^2}{2\pi} + \mathcal{O}(g^3)$. The quantity $y$ is the vortex fugacity for $(d, N) = (2, 2)$, but otherwise its interpretation is unclear. The small parameters $\epsilon = d - 2$, $(N-2)$ and $y^2$ are assumed to be of the same order, while the neglected terms (indicated as dots) are of order $\epsilon^2$. Eq. (1) admit two families of fixed point solutions parametrized by $\Delta = \epsilon \pi/2 - (N-2)f(\pi/2)$:

$$y_I^2 = \mathcal{O}(\epsilon^2), \quad f(g_I) = g_I[\epsilon/(N-2) + \mathcal{O}(\epsilon)], \tag{2}$$

and

$$y_{II}^2 = \Delta/(4\pi^3) + \mathcal{O}(\epsilon^2), \quad g_{II} = \pi/2 + \mathcal{O}(\epsilon). \tag{3}$$

When $\epsilon/(N-2) \to 0$ one recovers from $(y_I, g_I)$ the fixed-point of Ref. [12], while for $N = 2$ and $\epsilon = 0$ $(y_{II}, g_{II})$ goes into the Kosterlitz-Thouless fixed-point. As argued by C-H, the sign of $\Delta$ determines which fixed point governs the second-order transition:

- for $\Delta < 0$ the first FP is critical and the second is located outside the real domain;

- for $\Delta = 0$ the two solutions intersect;

- for $\Delta > 0$ the first FP is tricritical and the second is critical.

The collision of fixed point families is the source of the expected nonanalyticity of the critical exponents and defines the condition for the occurrence of the C-H line. Additionally, upon crossing the C-H line ($\Delta = 0$) the first fixed point changes its stability, which requires that the subdominant RG eigenvalue $e_2$ vanishes upon the collision. This constitutes a testable prediction which we aim to validate.

It is important to note, that the Kosterlitz-Thouless RG equations, employed in this analysis, are derived within the low-temperature expansion. Therefore the Eq. (1) are expanded not only in $\epsilon$, $(N-2)$ and $y^2$ but also $g$. It is well conceivable, that the higher-order terms might smoothen out the transition between the two families of solutions, with the nonanalyticity surviving only when $\epsilon = 0$. Our present study indicates a smooth crossover between the two asymptotic regimes, sharpening into a singularity only for $(d, N) \to (2, 2)$ and smoothening rapidly for increasing $d$. It also gives a hint on the actual shape of this crossover line in the $(d, N)$ plane. We find no indication of $e_2$ vanishing for any $N$ and $d \gtrsim 2.2$, where our calculation of this quantity may be considered as fully reliable. Quite contrary, $e_2$ remains well-separated from zero in the entire scanned region of the $(d, N)$ plane.

## 3   Functional RG and the derivative expansion

With the problem presented above in mind, we employ the one particle-irreducible variant of nonperturbative RG, adopting the exact Wetterich equation [13]

$$\partial_k \Gamma_k[\phi] = \frac{1}{2} \mathrm{Tr} \left\{ \partial_k R_k \left[ \Gamma_k^{(2)}[\phi] + R_k \right]^{-1} \right\},$$
(4)

as the point of departure. Eq. (4) describes the flow of the regularized effective action $\Gamma_k[\phi]$ upon varying the (momentum) cutoff parameter $k$ between the microscopic scale ($k = \Lambda$) and $k \to 0$. The quantity $\Gamma_k[\phi]$ evolves from the microscopic action $\Gamma_{k=\Lambda}[\phi] = \mathcal{S}[\phi]$ towards the free energy $\Gamma_{k\to 0}[\phi] = \mathcal{F}[\phi]$ as the infrared cutoff is gradually removed. The latter is implemented by adding a momentum-dependent function $R_k$ to the inverse propagator, which leads to damping of modes with momentum $q < k$ (while leaving the modes with $q > k$ unaffected). The trace in Eq. (4) sums over momentum and components of the order-parameter field $\phi$, while $\Gamma_k^{(2)}[\phi]$ denotes the second (functional) field derivative of $\Gamma_k[\phi]$.

The general framework resting upon Eq. (4) was successfully applied in a diversity of contexts over the last years (for reviews see e.g. [14–19]). The present study focuses on the canonical case of the $O(N)$ models, where the microscopic action is given by

$$\mathcal{S}[\phi] = \int d^d x \left[ \frac{1}{2} (\nabla \phi)^2 + \frac{\lambda}{8} \left( \phi^2 - \phi_0^2 \right)^2 \right].$$
(5)

Above we restricted to a form valid in the symmetry-broken phase, where the RG flow must be initiated in order to converge for $k \to 0$ to a fixed point describing the critical state. Note that $\phi$ is an $N$-component (real) field. The scheme of the derivative expansion proposes an ansatz

for the flowing effective action $\Gamma_k[\phi]$, classifying the (symmetry-allowed) terms according to the number of occurring derivatives and truncating terms of order higher than a prescribed value. In the present study we consider the $\partial^2$ truncation, where $\Gamma_k[\phi]$ is parametrized as

$$\Gamma_k[\phi] = \int d^d x \left\{ U_k(\rho) + \frac{1}{2}(Z_k(\rho) - 2\rho Y_k(\rho))(\nabla\phi)^2 + \frac{1}{4}Y_k(\rho)(\nabla\rho)^2 \right\}, \qquad (6)$$

retaining all the terms involving at most 2 derivatives and truncating those of higher order. Here $\rho = \frac{1}{2}\phi^2$, and the presence of the $Y_k(\rho)$ term distinguishes between the gradient coefficients of the longitudinal and transverse modes. Note that our convention differs from the most standard one (see e.g. Refs. [14, 19, 20]) by subtraction of the term $2\rho Y_k(\rho)$ in the $(\nabla\phi)^2$ coefficient. No truncation of the field dependencies is imposed, so that the set of three flowing functions $\mathcal{F}_k(\rho) = \{U_k(\rho), Z_k(\rho), Y_k(\rho)\}$ is determined by the flow itself and is not constrained by any pre-imposed parameterization. The procedure of projecting the Wetterich equation on the flow of $\mathcal{F}_k(\rho)$ amounts in essence to plugging Eq. (6) into Eq. (4) and is well described in literature (see e.g. Ref. [19]). The resulting flow equations are given in the Appendix. We note at this point that the longitudinal inverse propagator reads

$$\Gamma_\sigma^{(2)}(q, \rho) = Z_k(\rho)q^2 + U_k'(\rho) + 2\rho U_k''(\rho), \qquad (7)$$

while the transverse component of the inverse propagator is evaluated as

$$\Gamma_\pi^{(2)}(q, \rho) = [Z_k(\rho) - 2\rho Y_k(\rho)]q^2 + U_k'(\rho). \qquad (8)$$

The resulting set of three coupled nonlinear partial-differential flow equations can be analyzed numerically. It is convenient to rephrase the flow equations using the dimensionless (rescaled) quantities $\tilde{\rho}, u_k(\tilde{\rho}), z_k(\tilde{\rho}), y_k(\tilde{\rho})$, where

$$\rho = Z_k^{-1}k^{d-2}\tilde{\rho}, \quad U_k(\tilde{\rho}) = k^d u_k(\tilde{\rho}), \quad Z_k(\tilde{\rho}) = Z_k z_k(\tilde{\rho}), \quad Y_k(\tilde{\rho}) = Z_k^2 k^{2-d} y_k(\tilde{\rho}). \qquad (9)$$

In terms of these, the fixed-point behavior at the critical point is manifest. The rescaling factor $Z_k$ is related to the flowing anomalous dimension via $\eta = -\frac{k}{Z_k}\partial_k Z_k$ and is defined by imposing the condition $z_k(\tilde{\rho}_\eta) = 1$ with $\tilde{\rho}_\eta$ arbitrary. Note that the flowing anomalous dimension is evaluated from the longitudinal component of $\Gamma^{(2)}$. This choice allows for an arbitrary value of $N$, including $N = 1$, where the transverse modes are absent. We have verified that the differences in our results (relating to the critical point) obtained with $\eta$ evaluated from the longitudinal or from the transverse directions are negligible. We additionally choose $\tilde{\rho}_\eta = 0$.

Our analysis of the RG equations implements a discretization of the $\tilde{\rho}$ grid and follows two complementary paths. On one hand we integrate the flow starting from the initial condition of Eq. (5) and tune the initial condition so that the flow converges to the fixed point for vanishing cutoff scale. On the other hand, we solve directly the fixed-point equations. The subsequent linearization around the obtained solution and diagonalization of the obtained matrix allows for identifying the $\nu^{-1}$ exponent as the leading (and only positive) eigenvalue. These two distinct methods lead to very similar results, the latter being significantly faster and, in our assessment, also more accurate. We have extensively tested the sensitivity of the obtained results on the applied method [stability matrix analysis vs integration of the flow] as well as parameters of the grid discretization and accuracy of the integration. Our results indicate that errors related to numerical inaccuracies are way smaller as compared to those due to the truncation, and may be disregarded for all practical purposes relevant here.

Even though the framework of the derivative expansion was applied over many years, two impressive advancements related directly to the pure $O(N)$ models took place only very recently. The first concerns the resolution of the multicritical fixed point structure, including

identification of nonperturbative fixed points in $d = 3$ that had never been found before [21]. The second relates to establishing the methodology of the DE as a high-precision computational approach, capable of providing in $d = 3$ estimates of the critical exponents with accuracy comparable to (or even better than) those delivered by Monte-Carlo simulations and perturbative approaches. This required [22] calculations at order $\partial^4$ of the DE. For the less complex case of Ising symmetry-breaking ($N = 1$) the computation was performed [20] even up to order $\partial^6$. In addition to numbers (including errorbars), these studies delivered insights pointing towards rapid convergence of the DE, emphasizing (and clarifying [23]) the role of the so-called principle of minimal sensitivity (PMS) [24]. The latter amounts to demanding that the analyzed quantity (e.g. a critical exponent) be (locally) stationary with respect to the regulator choice.

The present study utilizes the DE at order $\partial^2$, which, however, is entirely sufficient for the purposes described above. An extension of this study to the fourth-order DE would entail using 13 functions parametrizing the effective action (instead of 3) and would require a tremendous effort both analytical and numerical. The first calculations at the fourth order DE for $O(N)$ models were published only very recently [20] and were performed, so far, only in three spatial dimensions.

We emphasize that the employed framework is applicable in the entire $(d, N)$ plane which constitutes its unique advantage. We also note that a somewhat similar scan of the critical indices in the $(d, N)$ plane was performed in Ref. [25] using a simpler truncation of functional RG, where the field dependencies of $Z$ and $Y$ were dropped. For an analogous calculation restricted to $N = 1$ see Ref. [26]. The limit $d \to 2^+$ with $N > 1$ was also examined in Ref. [27] within another simplified functional RG truncation.

## 3.1 Regulator choice

In the numerical evaluation of the flow equations we implement the Wetterich cutoff [14]

$$R_k(q) = \alpha Z_k q^2 / [\exp(q^2/k^2) - 1],$$ 
(10)

with a variable parameter $\alpha$. Refs. [22, 23] reveal the increasing role of the PMS principle in high-precision evaluation of the critical indices upon elevating the truncation order. At the $\partial^2$ order of the DE this dependence is however relatively modest. In Fig. 2 we demonstrate the evolution of the PMS value of $\alpha$ varying dimensionality. We find a notable increase of variation of $\alpha_{PMS}$ approaching $d = 2$ and no PMS value in the immediate vicinity of $d = 2$. In $d = 3$ our results for $\eta$ and $\nu$ [e. g. $\eta_{PMS}(d = 3, N = 2) \approx 0.047$ and $\nu_{PMS}(d = 3, N = 2) \approx 0.67$] coincide (up to two digits) with those of Ref. [22] obtained at the $\partial^2$ truncation order with the same regulator [see Table XVI of Ref. [22]]. In principle a small difference in the obtained values might arise due to dropping the terms of order higher than 4 [arising from multiplying the functions $\Gamma^{(3)}$] in the calculation of Ref. [22]. This however turns out not to influence the obtained numbers up to the precision of two digits.

In Fig. 3 the PMS value of the exponent $\eta$ is compared to the value of $\eta$ obtained for $\alpha = 2$ as function of $d$. Except for the immediate vicinity of $d = 2$ the difference between the two cases is negligible. The subsequent illustration (Fig. 4) exhibits the dependence of $\eta$ on $\alpha$ for a sequence of dimensionalities very close to 2. In particular, it demonstrates that no PMS value of $\alpha$ could be found for $d$ very close to 2 (i.e. for $d < d_0 \approx 2.01$).

An alternative procedure of optimizing the regulator was introduced in Ref. [28] specifically for $(d, N) = (2, 2)$. For this case, when integrating the flow in the algebraic (low-$T$) phase one does not recover the expected line of fixed points exactly, but only it the form of slightly tilted plateaus (quasi-fixed points). The plateau slope can be positive or negative depending on $\alpha$. One may therefore tune $\alpha$ so that the quasi-fixed point becomes transformed into a true fixed point. This constitutes a phenomenological procedure of compensating the deficiency of

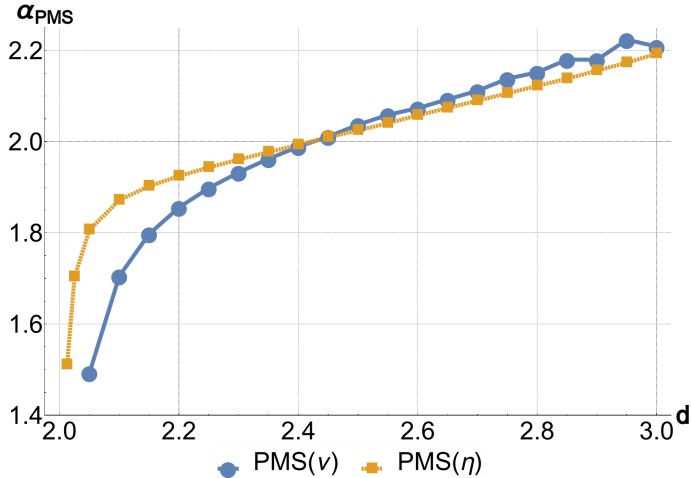

Figure 2: (Color online) Evolution of the PMS values of the regulator parameter $\alpha$ depending on dimensionality $d$ for $N = 2$. A substantial increase of its variation occurs for $d$ approaching 2. No PMS value is found for $d$ in the immediate vicinity of 2.

the truncation with a 'smart' regulator choice, which however enforces by hand the existence of the fixed-point line in the low-$T$ phase. For $T$ high enough (above the KT phase transition) the fixed point cannot be obtained for any value of $\alpha$, which signals the normal phase. Ref. [28] identified the 'optimal' value $\alpha_{opt} \approx 2.0$ at $T = T_{KT}$ and $\alpha_{opt}(T) < 2.0$ for $T < T_{KT}$. Note however, that this 'optimal' value (alike PMS) does depend on the renormalization point $\tilde{\rho}_\eta$. We also point out that if the procedure of regulator tuning is abandoned, the KT transition is captured [29, 30] in a form of an extremely sharp crossover into a phase characterized by an enormously large (but finite) correlation length which would be practically indistinguishible from infinite in an experiment or simulation.

In what follows we present our results for $\eta(d, N)$ and $\nu^{-1}(d, N)$ as obtained keeping the regulator fixed, with $\alpha = 2.0$. On one hand, this corresponds to an 'average' value for $d \in (2, 3]$ (at least for $N = 2$), on the other it is close to the 'optimal' value for $(d, N) = (2, 2)$. We emphasize that the differences between the PMS values of critical exponents (whenever $\alpha_{PMS}$ can be identified) and the values obtained at $\alpha = 2$ are relatively small. We also verified that the key results of the paper (see Sec. 5) are not changed if the PMS regulators are used (whenever they exist, i.e. for $d$ sufficiently separated from 2).

## 4 Dimensionality $d = 2$

In this section we analyze the case $d = 2$, approaching $N = 2$ from below. For the KT transition $[(d, N) = (2, 2)]$, the (complete) DE at order $\partial^2$ was addressed in Refs. [28, 29, 31]. The flow equations solved in the present paper are equivalent to those analyzed therein at the fixed point. For studies of the KT transition with other truncations of the functional RG, see Refs. [30, 32–38]. We point out that the present approach, despite the lack of vortices present as explicit degrees of freedom, accurately reproduces the key features of the KT transition, including the phase stiffness jump, the value of $\eta$ and the essential singularity of the correlation length.

The values of the critical exponents $\nu^{-1}$ and $\eta$ are however exactly known also for

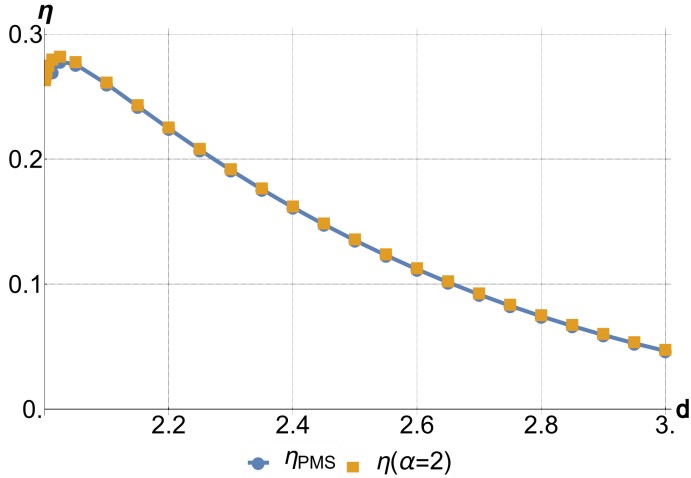

Figure 3: (Color online) Comparison of the exponent $\eta$ calculated for $\alpha_{\mathrm{PMS}}$ and $\alpha = 2$ depending on dimensionality $d$ for $N = 2$.

$(d = 2, N < 2)$ [39], providing a suitable opportunity for further benchmarking our results. For $t$ defined by $N = -2\cos(2\pi/t)$, $t \in [1, 2]$ the exact critical exponents read [39]

$$\nu^{-1} = 4 - 2t\,, \qquad \eta = 2 - t/2 - 3/(2t)\,, \tag{11}$$

and, for $N = 2 - \delta$, can be expanded in $\delta$ as follows:

$$\nu^{-1} = \frac{4}{\pi}\sqrt{\delta} + O(\delta)\,, \qquad \eta = \frac{1}{4} + \frac{1}{4\pi}\sqrt{\delta} + O(\delta)\,. \tag{12}$$

The first derivatives of both the exponents with respect to $N$ diverge as $N \to 2^-$, providing a clear indication of nonanalyticity of $\nu^{-1}(d, N)$ and $\eta(d, N)$ at $(d, N) = (2, 2)$.

Fig. 5 shows a comparison between the results obtained by us within the present functional RG truncation and the exact values of the critical exponents. The second-order DE approach yields systematically overestimated values of $\eta$ and fairly accurate values of $\nu^{-1}$. More importantly, our results capture the nonanalytical behavior of the exponents in the vicinity of $N = 2$. A power-law fit for $\nu^{-1}(d = 2, N)$ in the neighborhood of $N = 2$ yields the exponent 0.45, which is relatively close to the exact value. We note that our results slowly oscillate around the prediction of Nienhuis; we underestimate $\nu^{-1}$ very close to $N = 2$, and overestimate it for lower $N$. As concerns $\eta(d = 2, N)$ in the vicinity of $N = 2$, a power-law fit yields the exponent 0.77, which is clearly overestimated as compared to the exact value $1/2$.

We attribute the inaccuracies concerning the numerical values of the exponents to the low level of the implemented truncation and point out that the case of $d = 2$ is the least favorable for the present approach due to relatively large values of the anomalous dimension. [19, 20, 22, 23] The accuracy of our method is expected to increase upon raising $d$. It is nonetheless doubtless from our above results that the second-order DE is able to capture nonanalytic behavior of the critical exponents at $(d, N) = (2, 2)$. In the following section we use an analogous strategy in an attempt to identify the nonanalyticities at $d > 2$ expected to occur along the C-H line.

We note at this point that the C-H mechanism may be interpreted as a change of the universality class of the transition caused by a change of relevance of vortices. These may be neglected above the C-H line, but become important below it. One might wonder if the present approach does not suppress the vortices and in consequence is not really adequate to address the posed problem. For $(d, N) = (2, 2)$ the vortex-dominated picture is described

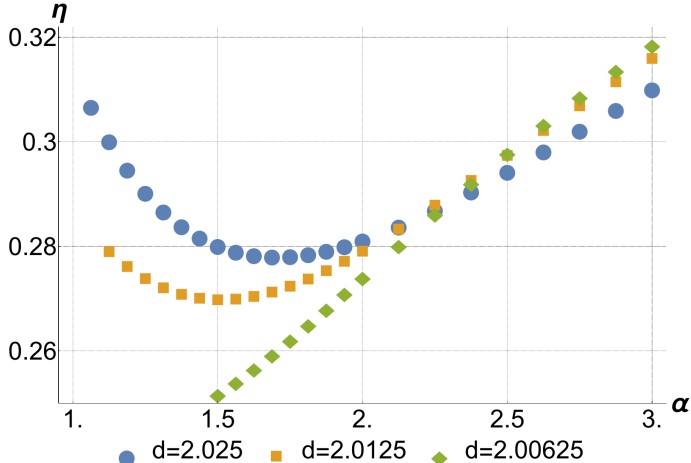

Figure 4: (Color online) Variation of $\eta$ depending on $\alpha$ for a sequence of dimensionalities in close vicinity of 2 and $N = 2$. No PMS value is found for $d$ sufficiently low.

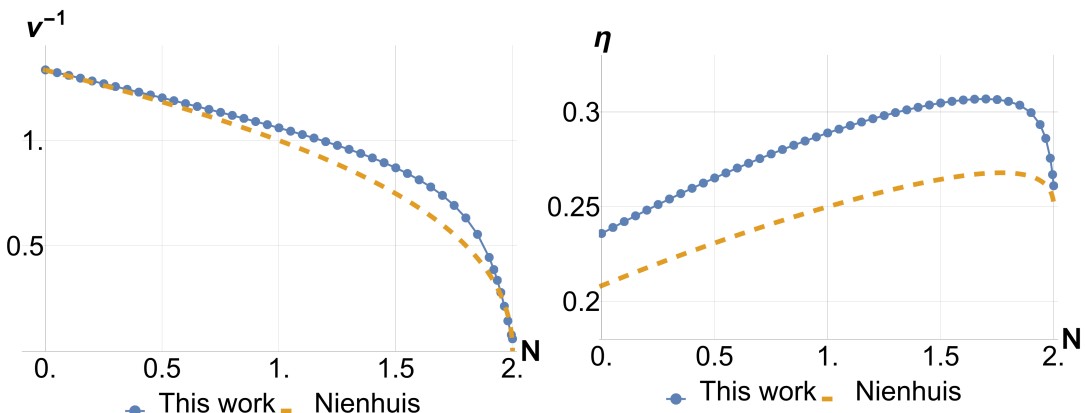

Figure 5: The critical exponents $\nu^{-1}$ and $\eta$ as functions of $N$ for $d = 2$. The singular first derivatives at $N = 2$ are clearly visible.

accurately for a fine-tuned regulator, however as already mentioned, if an arbitrary regulator is implemented, the KT transition is also captured in the form of an extremely sharp crossover. Importantly, as demonstrated by the work of Motrunich-Vishwanath [40] (see also Ref. [41]), vortex-like excitations are also relevant for the Heisenberg ($N = 3$) transition at dimensionality $d = 3$. The authors of this study addressed the nature of the phase transition in the $O(3)$ sigma model where they (artificially) suppressed vortices. They obtained a phase transition from a completely different universality class (characterized in particular by a very large anomalous dimension $\eta \approx 0.6$). There is no doubt that the transition obtained by us is in the Heisenberg (and not the non-compact CP universality class discussed in Ref. [40]). The relevant excitations are therefore captured. The onset of vortex-dominated physics across the C-H line is also evident from inspecting the profiles of the functional fixed point solutions which we present in Sec. 6.

## 5 The Cardy-Hamber line

In an attempt to detect the C-H line, we identify a (functional) fixed point corresponding to $(d, N)$ located far away from the expected nonanalyticity. This can be done by integrating the flow (tuning the initial condition so that the system flows sufficiently close to a fixed-point solution). We subsequently study the evolution of $v^{-1}$ and $\eta$ as either $d$ or $N$ varies towards the region where the C-H line should be found. In practice we either gradually decrease $d$ or increase $N$. The fixed point at $(d, N)$ serves as the initial condition for the fixed-point equations at $(d - \delta d, N)$ or $(d, N + \delta N)$, which (after discretization) are solved using standard algebraic routines. We are able to scan the $(d, N)$ plane and extract numerically the functions $\eta(d, N)$ and $v^{-1}(d, N)$ traversing the region where the C-H line is expected.

In the following subsections we present the results of this scanning procedure along horizontal (subsection 5.1) and vertical (subsections 5.2 and 5.3) trajectories in the $(d, N)$ plane. We note that the procedure of finding the fixed point becomes progressively harder when lowering $d$ and the step in the $(d, N)$ plane must then be tiny. This is (at least partially) related to the fact that the profile of the fixed point effective potential acquires at $d$ low an increasingly strong variation at large $\tilde{\rho}$. For selected choices of $(d, N)$ we checked the results against those obtained by integration of the flow. We note that for $N > 2$ we were not able to solve the fixed-point equations for $d$ arbitrarily close to 2, but anyway significantly lower than the anticipated position of the C-H line.

### 5.1 $d$-dependence

The left panel of Fig. 6 demonstrates the dependence of the $v^{-1}(d)$ exponent on dimensionality for a sequence of values of $N$. Our results are juxtaposed with the known exact results $v^{-1}(d, N = \infty) = d - 2$. In the limit $d \to 2^+$, the exponent $v^{-1}(d, N = 2)$ vanishes with a very large (presumably infinite) derivative. Only at this point are we dealing with a clear nonanalyticity of $v^{-1}$. For each $N > 2$, there exists a characteristic value of dimensionality $d_c(N)$ at which $v^{-1}$ converges rapidly towards the large-$N$ behavior; $d_c(N)$ increases for growing $N$.

Our results for the exponent $\eta(d)$ as a function of the dimensionality are presented in the right panel of Fig. 6 along with the exact result $\eta(d, N = \infty) = 0$. The distinct characteristic of the case $N = 2$ is equally pronounced as for the exponent $v^{-1}$. While $\eta(d, N = 2)$ approaches a non-zero value $\eta(d = 2, N = 2) \approx 0.27$ in the limit $d \to 2$, the curves corresponding to $N > 2$ converge towards 0 in agreement with the $\epsilon$-expansion results [42]. As we already remarked, we are not able to get arbitrarily close to $d = 2$ for $N > 2$, however the range of $d$ where the curves in Fig. 6 terminate is significantly lower that the expected position of the C-H line. The dimensionality $d_c(N)$ corresponds to the maximum of $\eta(d)$, where one crosses over between the large-$N$-like and small-$N$-like behaviors.

The difference between the behavior of the critical exponents between low-$N$ and large-$N$ regimes fits nicely into the picture presented by Cardy and Hamber and it is natural to relate $d_c(N)$ with the C-H line. We also note that $d_c(N)$ is situated close to the predicted position of the C-H line. However, the crossover from low-$N$-like to large-$N$-like behavior remains analytical (or at least of the $C^2$ type). This suggests that the fixed points' collision described in Ref. [10] is actually avoided within our calculation. Instead, the obtained picture indicates a crossover between the situations controlled by the two fixed points of the C-H analysis with no indication of nonanalyticity [except for the immediate vicinity of $(d, N) = (2, 2)$].

### 5.2 $N$-dependence

The picture becomes even more transparent when we inspect the $N$-dependence of the critical exponents. The left panel of Fig. 7 illustrates the variation of $v^{-1}$ between $N = 1$ and $N = 6$



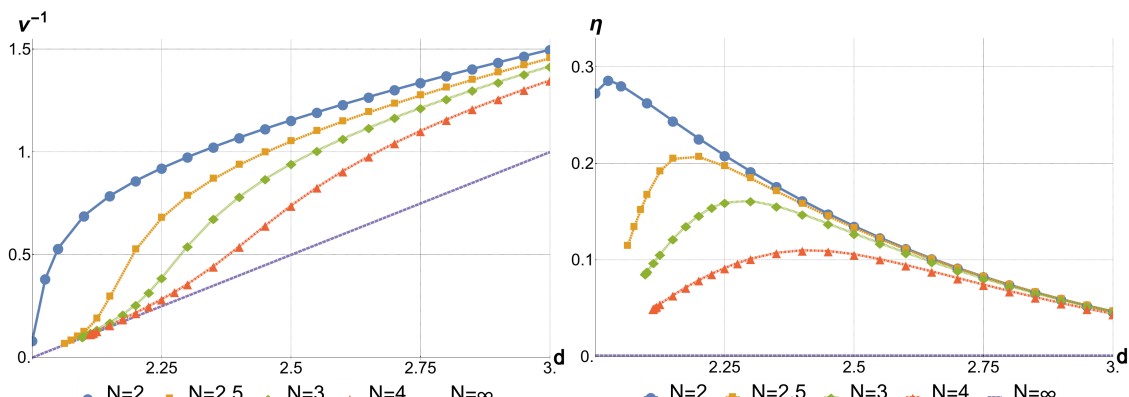

Figure 6: The critical exponents $\nu^{-1}$ and $\eta$ as functions of $d$ for a sequence of values of $N$. In particular, for $N = 2$ we find $\eta(d \to 2^+, N = 2)$ approaching the value $\approx 0.27$ with a singular first derivative.

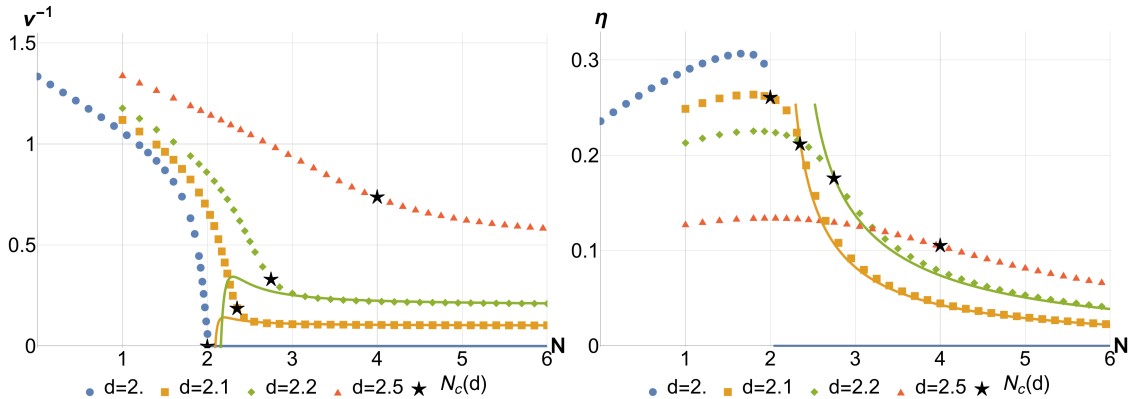

Figure 7: The critical exponents $\nu^{-1}$ and $\eta$ as functions of $N$ for a sequence of values of $d$. Continuous lines denote the $\epsilon$-expansion predictions; the lines corresponding to $d = 2.5$ were removed to avoid obscuring the illustration. The stars indicate our estimate of $N_c(d)$ (see the main text).

for a sequence of values of $d$. These results are compared to the predictions of the $\epsilon$-expansion at order $\epsilon^4$ [42]. In two dimensions, $\nu^{-1}$ approaches 0 in a square-root like fashion, exhibiting the nonanalyticity at $N = 2$. At higher dimensions $\nu^{-1}$ reaches the large-$N$ limit, but no clear nonanalyticity is present. Instead, a crossover-like behavior between low-$N$ and large-$N$ regimes occurs. This crossover smoothens progressively upon increasing $d$. This transition seems to be closely related to the point where the divergence between our results and the predictions of the $\epsilon$-expansion occurs.

The right panel of Fig. 7 displays a comparison between our results for the exponent $\eta$ and the predictions of the $\epsilon$-expansion. At the point $(d, N) = (2, 2)$, $\eta$ has a discontinuity and a singular derivative. These two properties do not survive when we move to larger dimensions; $\eta(d > 2, N)$ slowly approaches its large-$N$ limit ($\eta(d, N = \infty) = 0$) in an apparently analytical fashion. Our results indicate that the nonanalyticity of the critical exponents present at $(d, N) = (2, 2)$ becomes smoothened as we move to higher dimensions.

We finally examine the derivatives of the critical exponents $\nu^{-1}(d, N)$ and $\eta(d, N)$. In Fig. 8 we plot $\partial_N^2 \nu^{-1}$ and $\partial_N \eta$; by following their maxima/minima we observe the emergence of the

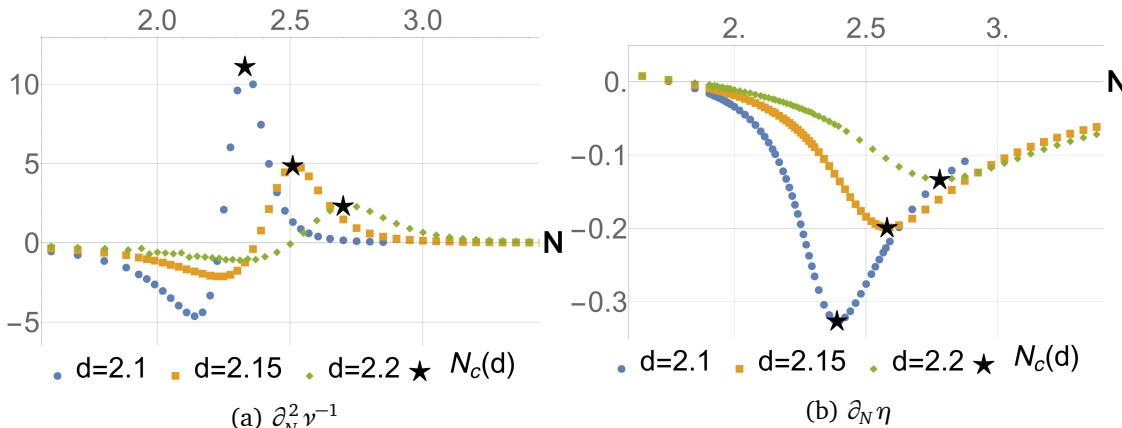

(a) $\partial_N^2 v^{-1}$        (b) $\partial_N \eta$

Figure 8: The derivatives of the critical exponents as functions of $N$ for a sequence of values of $d$. The stars indicate the maxima/minima, which serve as our defining property of the $N_c(d)$ line.

singularities at $(d = 2, N = 2)$. The values of these functions at their extrema are increasingly large as $\epsilon = d - 2$ approaches 0, yet they become infinite only in this limit. We adopt the position of these extrema as the (phenomenological) property determining the position of the crossover line $N_c(d)$ between the large-$N$-like and the small-$N$-like regimes.

The maxima of $\partial_N^2 v^{-1}$ and the minima of $\partial_N \eta$ lie very close to each other. Notably, the values of the derivatives at maxima become increasingly small as $d$ grows signalling smoothening of the crossover between the large-$N$-like and the small-$N$-like behaviors. Between $d = 2.75$ and $d = 3$. the maximum of $\partial_N^2 v^{-1}$ disappears completely. The loci of the maxima of $\partial_N^2 v^{-1}$ and the minima of $\partial_N \eta$ are plotted in Fig. 9 in the $(d, N)$ plane. In the vicinity of $(d, N) = (2, 2)$, they are found close to the expected position of the Cardy-Hamber line.

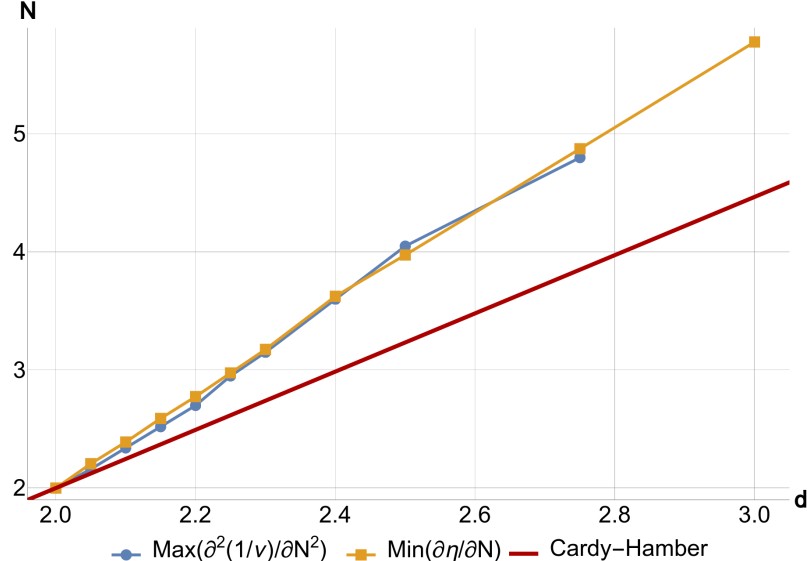

Figure 9: Loci of the maxima of $\partial_N^2 v^{-1}$ and the minima of $\partial_N \eta$ compared to the C-H line.

## 5.3 Subdominant eigenvalue

We now analyze the behavior of the subdominant RG eigenvalue $e_2$ which determines the correction to scaling exponent $\omega$. The fixed point collision scenario by C-H requires as a necessary condition that the subdominant eigenvalue vanishes identically everywhere on the C-H line. Fig. 10 demonstrates the dependence of $e_2$ on $N$ for a sequence of values of $d$. Resolving the limit of $e_2$ as $(d, N) \to (2, 2)$ is not possible at the present approximation level and would require a substantial refinement of the truncation. Our results for $e_2$ are however fully reliable in the vicinity of $d = 3$. For this case in Fig. 10 they are juxtaposed with the very accurate values obtained within the derivative expansion at order $\partial^4$ [22] and the results of the $1/N$ expansion [43]. We note that the presented $\partial^4$ results are of similar accuracy as the most recent Monte Carlo results, and were chosen as a convenient reference point offering data for a wide range of $N$. The differences between the values obtained at order $\partial^2$ and $\partial^4$ turn out negligible for the present purposes; note that the more precise results are separated from 0 even a bit further than ours.

We observe a continuous interpolation between the well-established results. The curve for $d = 3$ remains very well separated from zero in the entire range of $N$. Note that consistency with the C-H scenario would require $e_2$ to be located in a completely different range of values. The obtained picture provides strong evidence against the existence of the C-H line for $d = 3$. Upon reducing dimensionality, the picture evolves continuously and remains qualitatively unchanged down to $d \approx 2.2$. In an exact calculation, in the limit $d \to 2^+$ we would expect a violent increase of the maximal value of $e_2$ towards zero, such that $e_2(d \to 2^+, N = 2) \to 0^-$. This feature is not captured at the present level of approximation. Nonetheless the results of this section allow us to exclude the possibility of existence of the C-H line of nonanalyticities in a broad vicinity of $d = 3$. In consequence, the validity of the C-H fixed-point collision prediction, which is built upon the $2 + \epsilon$ expansion, would require that either the nonanalyticity line terminates at some point close to $d = 2$, or it extends up to large $N$, but becomes vertical at some dimensionality close to $d = 2$. The question concerning the mechanism governing the

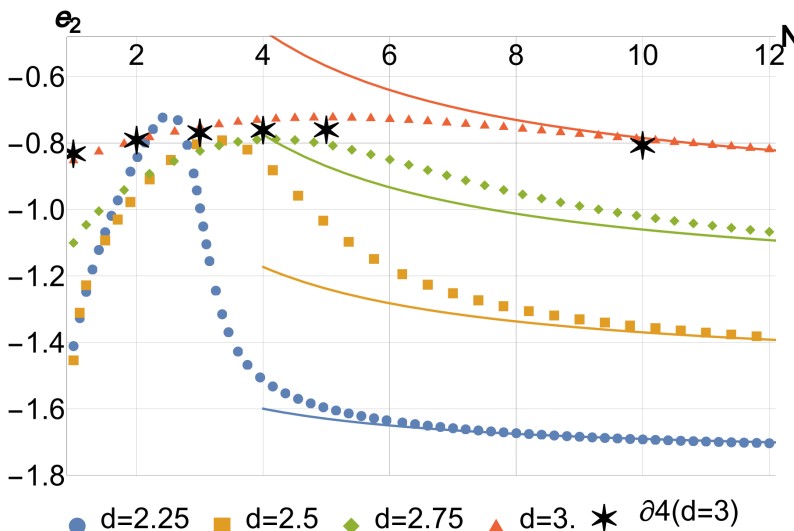

Figure 10: The subdominant eigenvalue $e_2$ as function of $N$ for a sequence of values of $d$. Continuous lines denote the $1/N$-expansion predictions. The stars denote the results of derivative expansion at order $\partial^4$ for $d = 3$.

change of the vortices' relevance without vanishing of $e_2$ requires clarification. In particular, as argued in Ref. [44] the $2 + \epsilon$ expansion continued to $d = 3$ is expected to describe the $CP^1$ model [40] [which is very distinct from the $O(3)$ model]. The failure of the $2 + \epsilon$ expansion

to account for the $O(3)$ model in $d = 3$ fits very nicely into the $C - H$ scenario. Intriguingly the $2 + \epsilon$ predictions deviate from our results below the crossover line obtained by us at low $d$ (compare Fig. 7). A resolution of this puzzling issue is not achieved in the present work and calls for further investigations.

# 6 Fixed points

We now inspect the fixed-point profiles and investigate how the onset of the vortex-dominated physics upon increasing $d$ (or reducing $N$) is reflected by their violent change in the vicinity of the C-H (crossover) line.

Fig. 11 demonstrates the variation of the functional fixed point obtained by us across the $(d, N)$ plane. In large dimensions, the fixed point effective potential very much resembles the effective potential of the $\phi^4$ theory, at least up to $\tilde{\rho}$ corresponding to the minimum. The derivative of the effective potential is almost exactly linear in $\tilde{\rho}$. In addition, there is almost no difference between the longitudinal and transverse fluctuation suppressors [$z_\sigma(\tilde{\rho}) = z_k(\tilde{\rho})$ and $z_\pi(\tilde{\rho}) = z_k(\tilde{\rho}) - 2\tilde{\rho} y_k(\tilde{\rho})$ respectively]. Both fluctuation suppressors are almost constant as functions of $\tilde{\rho}$.

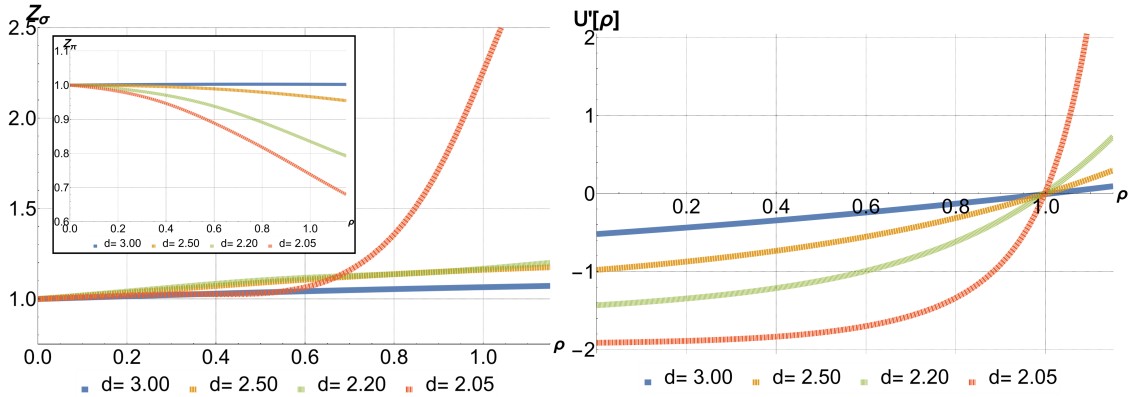

Figure 11: Critical fixed points for $N = 2.5$ and a series of values of $d$. The left panel shows the fluctuations suppressors: longitudinal $z_\sigma(\tilde{\rho})$ (main plot) and transverse $z_\pi(\tilde{\rho})$ (inset). Particularly visible is the drastic (but smooth) increase of $z_\sigma(\tilde{\rho})$ upon crossing the C-H line located slightly below $d = 2.2$. The right panel shows the derivative of the local potential $u'(\tilde{\rho})$. The axes were rescaled, so that the minimum of the local potential always lies at $\tilde{\rho} = 1$.

When lowering the dimensionality, the fixed point effective potential acquires a pronounced minimum, effectively trapping the order-parameter in its close vicinity. At the same time, the longitudinal fluctuations become strongly suppressed while the cost of the transverse fluctuation decreases. A violent (but smooth) increase of $z_\sigma(\tilde{\rho})$ appears upon traversing the vicinity of the C-H crossover line and continues rapidly as the dimension $d$ decreases. This structure of the effective action, with (almost) fully suppressed longitudinal fluctuations, resembles the non-linear $\sigma$ model. The fixed point structure present in low dimensions (strong transverse and weak longitudinal fluctuations) is consistent with the prediction that vortices change relevance upon traversing the crossover line.

In Fig. 12 we also exhibit the evolution of the fixed point parameters $u''(\tilde{\rho})$, $z_\sigma(\tilde{\rho})$ and $z_\pi(\tilde{\rho})$ evaluated at the minimum of the local potential $\tilde{\rho} = \tilde{\rho}_0$ upon varying $d$ and $N$. The dependence of these parameters on $d$ is non-trivial, in some cases showing more than one local

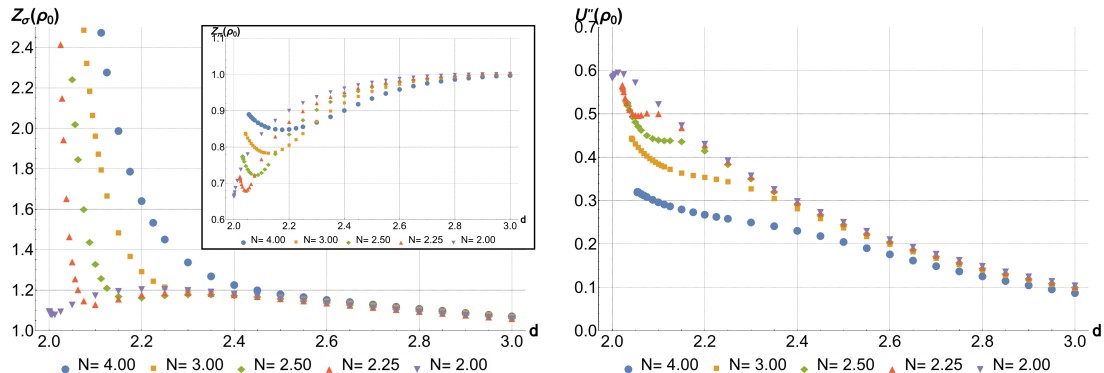

Figure 12: Critical fixed point parameters at the minimum $\tilde{\rho}_0$ of the local potential as a function of $d$ and $N$. The left panel shows the fluctuations suppressors: longitudinal $z_\sigma(\tilde{\rho}_0)$ (main plot) and transverse $z_\pi(\tilde{\rho}_0)$ (inset). The right panel shows the second derivative of the local potential $u''(\tilde{\rho}_0)$.

extremum. Our earlier phenomenological identification of the position of the C-H line via the extrema of the derivative of the critical exponents (see Sec. 5) turns out to lie very closely to the maxima of $u''(\tilde{\rho}_0)$, the minima of $z_\sigma(\tilde{\rho}_0)$, as well as the inflection points of $z_\pi(\tilde{\rho}_0)$.

The analysis of the fixed point profiles can also serve to illustrate two problems arising in the numerical analysis of the functional RG equations close to $d = 2$. For every value of $N > 2$, both $z_\sigma(\tilde{\rho})$ and $u''(\tilde{\rho})$ exhibit very large derivatives for $\tilde{\rho}$ large (beyond the local potential minimum), which seem to diverge as we approach the limit $d \to 2^+$. This divergences make our numerical procedure of approximating the derivatives with finite differences unreliable.

The second problem lies in the numerical calculation of the loop integrals. For low values of $u'(0) \gtrsim -\alpha$, the transverse propagators, present in our calculation suffer from a pole close to $q^2 \approx -\alpha - u'(0)$, where $q$ is the loop-integral momentum. Even though in the studied cases this pole lies on the imaginary axis, it can strongly affect the precision of numerical integration when $u'(0)$ is sufficiently close to $-\alpha$; this again happens for any $N > 2$ when $d \to 2^+$. Due to the above technical problems, throughout the paper we have removed the obtained fixed points for which sufficient numerical precision could not be achieved.

The above analysis of the fixed point structure strongly indicates that the fixed point effective action smoothly interpolates between the behavior characterized by the $\phi^4$ theory in large dimensions and the nonlinear-$\sigma$ model in low dimensions. The cross-over is smooth, occurs in the vicinity of the predicted C-H line, and may be interpreted as being due to the vortices becoming a relevant perturbation.

# 7 Conclusion

In this paper we have addressed the analyticity of the critical exponents $\nu(d, N)$ and $\eta(d, N)$ of the $O(N)$ models in $d \geq 2$, $N \geq 2$, varying continuously dimensionality $d$ and the number of order parameter components $N$. We confronted our results obtained from functional renormalization group (truncated at order $\partial^2$ of the derivative expansion) against those derived long ago by Cardy and Hamber within the $2 + \epsilon$ expansion [10] of the non-linear sigma model at order $\epsilon$. Except for $(d, N) = (2, 2)$ we did not recover signatures of nonanalyticities of the critical exponents that would be manifest from the properties of the first two derivatives of the functions $\nu^{-1}(d, N)$ and $\eta(d, N)$. Instead, we obtained a locus of maxima of the derivatives of the functions $\nu^{-1}(d, N)$ and $\eta(d, N)$, terminating with a singularity at $(d, N) = (2, 2)$ and

a related crossover of the critical exponents between large-$N$-like and small-$N$-like regimes. Moreover, our results indicate that the subdominant eigenvalue of the RG transformation does not vanish anywhere in the $(d, N)$ plane (except, perhaps at dimensionalities close to $d = 2$, where our approach is not sufficiently accurate) - which is a necessary condition for the fixed point collision yielding the nonanalytical critical exponents in the C-H scenario. Quite contrary, at least for $d \gtrsim 2.2$ where our calculation of this quantity is reliable, it remains very well separated from zero. This result demonstrates the non-existence of the C-H line in the vicinity of $d = 3$ and constitutes a strong disagreement with the results of Ref. [10].

We cannot rule out the possibility that the nonanalyticity line indeed exists in a narrow strip around $d = 2$, but on the other hand we note that the prediction of Cardy-Hamber involves the phenomenon of fixed-point collision, which may be smoothened when terms of higher order in $\epsilon$ and $g$ are taken into account. It is also not unimaginable that the C-H nonanalyticity arises as an artifact of the procedure of merging the flow equations coming from two completely distinct calculations, where in addition one of the two flowing parameters lacks a clear physical meaning except for $(d, N) = (2, 2)$.

As concerns the vicinity of $d = 2$ we cannot completely exclude the possibility that the locus of derivatives' maxima obtained by us at the present truncation level (order $\partial^2$ of the derivative expansion) is in fact a 'fingerprint' of the Cardy-Hamber line, which would build up into a true nonanalyticity upon including higher-order terms of the derivative expansion, indicating that the present framework is insufficient to capture the rather subtle 'fixed point collision' phenomenon. This possibility seems unlikely to us, since, if this was really the case, this insufficiency would apply both to the vicinity of $(d, N) = (2, 2)$, as well as larger dimensions, where our approximation scheme is expected to become increasingly more reliable [20, 22] and where the C-H approach can by no means be treated as accurate.

In summary, our work excludes the possibility of the existence of the Cardy-Hamber line of nonanalyticities of the critical exponents in the form envisaged by the C-H scenario in spatial dimensionality $d$ approaching three. It also suggests that this line is in fact a crossover also for $d$ close to two, and evolves into a true singularity only in the limit $(d, N) \rightarrow (2, 2)$.

## Acknowledgments

We thank Bertrand Delamotte for his help at the initial stages of this project as well as reading the first version of the manuscript and very useful comments. We are grateful to Maxym Dudka, Nicolas Dupuis and Nicolas Wschebor for discussions as well as valuable suggestions on the manuscript. We acknowledge support from the Polish National Science Center via grant 2017/26/E/ST3/00211.

## Appendix - fRG flow equations

In this section we present the RG equations that were used in this work. To simplify the expressions we first introduce the fluctuation suppressors: $Z_\sigma(\rho) = Z_k(\rho)$, $Z_\pi(\rho) = Z_k(\rho) - 2\rho Y_k(\rho)$ and the "dressed" propagators:

$$G_\sigma(\rho) = \left( Z_\sigma(\rho)q^2 + U'(\rho) + 2\rho U''(\rho) + R(k, \boldsymbol{q}^2) \right)^{-1} , \tag{13}$$

$$G_\pi(\rho) = \left( Z_\pi(\rho)q^2 + U'(\rho) + R(k, \boldsymbol{q}^2) \right)^{-1} , \tag{14}$$

where the index $k$ denoting the running scale dependence was dropped for clarity. The flow equations for the dimensional functions $U'(\rho)$, $Z_\sigma(\rho)$ and $Z_\pi(\rho)$ read:

$$\partial_k U'(\rho) = -\frac{1}{2}\int_q R^{(1,0)}(k,\boldsymbol{q}^2)\bigg((N-1)G_\pi(\rho)^2\big(\boldsymbol{q}^2 Z'_\pi(\rho)+U''(\rho)\big)$$

$$+ G_\sigma(\rho)^2\big(\boldsymbol{q}^2 Z'_\sigma(\rho)+3U''(\rho)+2\rho U^{(3)}(\rho)\big)\bigg), \tag{15}$$

$$\partial_k Z_\sigma(\rho) = \frac{1}{2d}\int_q R^{(1,0)}(k,\boldsymbol{q}^2)\bigg\{-4(N-1)\rho G_\pi(\rho)^5\big(R^{(0,1)}(k,\boldsymbol{q}^2)+Z_\pi(\rho)\big)\big(\boldsymbol{q}^2 Z'_\pi(\rho)+U''(\rho)\big)^2$$

$$\big(\boldsymbol{q}^2\big((d-4)Z_\pi(\rho)-4R^{(0,1)}(k,\boldsymbol{q}^2)\big)+dR(k,\boldsymbol{q}^2)+dU'(\rho)\big)$$

$$- G_\sigma(\rho)^2\Big[4\rho G_\sigma(\rho)^2\big(\boldsymbol{q}^2 Z'_\sigma(\rho)+3U''(\rho)+2\rho U^{(3)}(\rho)\big)$$

$$\Big(\big(R^{(0,1)}(k,\boldsymbol{q}^2)+Z_\sigma(\rho)\big)\big((d+4)\boldsymbol{q}^2 Z'_\sigma(\rho)+3dU''(\rho)+2d\rho U^{(3)}(\rho)\big)$$

$$+2\boldsymbol{q}^2 R^{(0,2)}(k,\boldsymbol{q}^2)\big(\boldsymbol{q}^2 Z'_\sigma(\rho)+3U''(\rho)+2\rho U^{(3)}(\rho)\big)\Big)$$

$$-16\boldsymbol{q}^2\rho G_\sigma(\rho)^3\big(R^{(0,1)}(k,\boldsymbol{q}^2)+Z_\sigma(\rho)\big)^2\big(\boldsymbol{q}^2 Z'_\sigma(\rho)+3U''(\rho)+2\rho U^{(3)}(\rho)\big)^2$$

$$-4\rho G_\sigma(\rho)Z'_\sigma(\rho)\big((2d+1)\boldsymbol{q}^2 Z'_\sigma(\rho)+6dU''(\rho)+4d\rho U^{(3)}(\rho)\big)$$

$$+dZ'_\sigma(\rho)+2d\rho Z''_\sigma(\rho)\Big]$$

$$+8(N-1)\boldsymbol{q}^2\rho G_\pi(\rho)^4\big(\boldsymbol{q}^2 Z'_\pi(\rho)+U''(\rho)\big)$$

$$\big(-R^{(0,2)}(k,\boldsymbol{q}^2)\big(\boldsymbol{q}^2 Z'_\pi(\rho)+U''(\rho)\big)-2Z'_\pi(\rho)\big(R^{(0,1)}(k,\boldsymbol{q}^2)+Z_\pi(\rho)\big)\big)$$

$$+4d(N-1)G_\pi(\rho)^3\bigg(\frac{\boldsymbol{q}^2\rho Z'_\pi(\rho)^2}{d}+\big(Z_\sigma(\rho)-Z_\pi(\rho)\big)\big(\boldsymbol{q}^2 Z'_\pi(\rho)+U''(\rho)\big)\bigg)$$

$$-\frac{d(N-1)G_\pi(\rho)^2\big(\rho Z'_\sigma(\rho)-Z_\sigma(\rho)+Z_\pi(\rho)\big)}{\rho}\bigg\}, \tag{16}$$

$$\partial_k Z_\pi(\rho) = \frac{1}{2d}\int_q R^{(1,0)}(k,\boldsymbol{q}^2)\bigg\{-\frac{1}{\rho}\Big[G_\pi(\rho)^2\Big(G_\sigma(\rho)^2\big(d\big(\boldsymbol{q}^2(Z_\sigma(\rho)-Z_\pi(\rho))+2\rho U''(\rho)\big)^2$$

$$\big(R^{(0,1)}(k,\boldsymbol{q}^2)+Z_\sigma(\rho)\big)+8\boldsymbol{q}^2(Z_\pi(\rho)-Z_\sigma(\rho))\big(\boldsymbol{q}^2(Z_\pi(\rho)-Z_\sigma(\rho))-2\rho U''(\rho)\big)$$

$$\big(R^{(0,1)}(k,\boldsymbol{q}^2)+Z_\sigma(\rho)-\rho Z'_\pi(\rho)\big)+4R^{(0,2)}(k,\boldsymbol{q}^2)\big(\boldsymbol{q}^3(Z_\sigma(\rho)-Z_\pi(\rho))+2\rho q U''(\rho)\big)^2\big)$$

$$-4\boldsymbol{q}^2 G_\sigma(\rho)^3\big(\boldsymbol{q}^2(Z_\sigma(\rho)-Z_\pi(\rho))+2\rho U''(\rho)\big)^2\big(R^{(0,1)}(k,\boldsymbol{q}^2)+Z_\sigma(\rho)\big)^2$$

$$-2G_\sigma(\rho)\Big(d(Z_\pi(\rho)-Z_\sigma(\rho))\big(\boldsymbol{q}^2(Z_\pi(\rho)-Z_\sigma(\rho))-2\rho U''(\rho)\big)$$

$$+2\boldsymbol{q}^2\big(-Z_\sigma(\rho)+\rho Z'_\pi(\rho)+Z_\pi(\rho)\big)^2\Big)+d(N-1)\rho Z'_\pi(\rho)+dZ_\sigma(\rho)-dZ_\pi(\rho)\Big)\Big]$$

$$-\frac{1}{\rho}\Big(G_\pi(\rho)^3 G_\sigma(\rho)^2\big(R^{(0,1)}(k,\boldsymbol{q}^2)+Z_\pi(\rho)\big)\big(\boldsymbol{q}^2(Z_\sigma(\rho)-Z_\pi(\rho))+2\rho U''(\rho)\big)^2$$

$$\big(\boldsymbol{q}^2\big((d-4)Z_\pi(\rho)-4R^{(0,1)}(k,\boldsymbol{q}^2)\big)+dR(k,\boldsymbol{q}^2)+dU'(\rho)\big)\Big)$$

$$+4G_\pi(\rho)G_\sigma(\rho)^2 Z'_\pi(\rho)\big(\boldsymbol{q}^2\big(dZ_\sigma(\rho)+\rho Z'_\pi(\rho)\big)$$

$$-d\boldsymbol{q}^2 Z_\pi(\rho)+2d\rho U''(\rho)\big)-dG_\sigma(\rho)^2\big(Z'_\pi(\rho)+2\rho Z''_\pi(\rho)\big)\bigg\}. \tag{17}$$

In the formulas above $\int_{\boldsymbol{q}} = \int \frac{d^d\boldsymbol{q}}{(2\pi)^d}$. The flow of $Y_k(\rho)$ can be simply recovered as:

$$\partial_k Y_k(\rho) = \frac{\partial_k Z_\sigma(\rho) - \partial_k Z_\pi(\rho)}{2\rho} \ . \tag{18}$$

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
