# Peer review of "Analyticity of critical exponents of the $O(N)$ models from nonperturbative renormalization"

_SciPost Physics, doi:SciPost Phys. 10, 134 (2021)_

## Round 3 · Referee Report · Slava Rychkov (Referee 1) · 2021-1-17

Weaknesses

Not sure the model in this paper has a chance to capture the Cardy-Hamber mechanism

Report

Cardy and Hamber back in 1980 argued that vortices may affect the transition in the $O(n)$ model for $d>2$. These are the same vortices which cause the BKT transition for $n=2$, $d=2$. Cardy and Hamber argued that for $n>2$ and $d$ very close to 2 the vortices are irrelevant, so that the standard $2+\epsilon$ expansion a la Brezin and Zinn-Justin (which does not take vortices into account) is applicable, while for larger $d$ vortices become relevant and change the universality class of the transition. Along the line in the $(n,d)$ plane where vortices become relevant, the critical exponents should be continuous but not analytic.

One caveat of the Cardy-Hamber paper is that they do not present a first-principle derivation of their RG equations (Ref. [10] in Cardy-Hamber promises a derivation but I’m not sure it appeared in print.) So a better understanding of their mechanism, and of the difference in the transition in systems with allowed and suppressed vertices, would be welcome. I’m not aware of much work in this direction. One relevant paper is Ref. https://arxiv.org/abs/cond-mat/0311222 which discussed the phase transition in the O(3) model in $d=3$ with suppressed vortices. On the other hand, there was much related work about phase transitions in the 3d gauge theories coupled to matter, with monopoles playing the role of vortices. In this case it is known that depending on the number of the matter species, monopoles may be relevant or irrelevant. These latter models are sometimes called $CP^{N-1}$ models because of their microscopic description, and the above-mentioned $O(3)$ model with suppressed vortices is also called “non-compact $CP^{N-1}$”.

The paper under review studies the phase transition in the $O(n)$ model for various $d$ and $n$ using the FRG. They do not observe any non-analyticity line, and interpret this as evidence against Cardy-Hamber. I am not sure however I agree with this interpretation. Indeed, to test the Cardy-Hamber mechanism, one should to begin with have a model which allows for vortices. On the contrary, their model is a continuous scalar field theory with $O(n)$ symmetry. Since the field is continuous, the vortices are totally suppressed for any $n$ and $d$. Therefore, their model has nothing to do with the Cardy-Hamber mechanism, analyticity being totally expected given the built-in suppression of vortices. If my interpretation is correct, the paper might be misleading.

  • validity: -
  • significance: -
  • originality: -
  • clarity: -
  • formatting: -
  • grammar: -

Author:  Andrzej Chlebicki  on 2021-04-01  [id 1340]

(in reply to Report 1 by Slava Rychkov on 2021-01-17)
Category:
answer to question
reply to objection

Dear Professor Rychkov, we would like to thank you for reviewing our paper. We give our answer to your points below: 1. Despite the effort we were not able to identify Ref.[10] from the Cardy-Hamber paper. 2. We are grateful for directing us towards the Motrunich-Vishwanath paper, which we found very useful. 3. The evidence against the interpretation proposed in your report is as follows: i) The framework of the DE does not introduce vortices as explicit degrees of freedom. This however does not imply that the corresponding physics is not captured. The K-T singularity in d=2 is accurately reproduced for a tuned regulator. For an arbitrary regulator, it appears in the form of an extremely sharp crossover between phases characterized by a small and an enormously large correlation length. For an arbitrary regulator, we obtain in the vicinity of d=N=2 a sharp crossover between a large-N and small-N behavior. This crossover might be interpreted as the C-H singularity 'smoothened' by the approximation. This interpretation however does not hold when moving away from d=2. If the C-H line exists and is recovered in the 'smoothened' form close to d=N=2, it should also be easily detected for d larger. There is not the slightest indication of it. ii) If the vortices (hedgehogs) were not captured in the vicinity of d=3, our approach would not yield the exponents of the Heisenberg universality class. This follows from the study of Motrunich-Vishwanath, who considered the O(3) model with suppressed vortices and obtained a phase transition very distinct from the O(3) Heisenberg transition. In particular, the Motrunich-Vishwanath transition is characterized by a very large anomalous dimension. There are no doubts that the predictions of the derivative expansion (DE) for d=3 are in the Heisenberg universality class (see Refs. 21, 22 for the most accurate estimates). It is also clear that the DE results are inconsistent with the Motrunich-Vishwanath predictions for the model with suppressed vortices. In consequence, the relevant excitations are captured by our approach. iii) The evolution between the regimes characterized by important and unimportant vortices is also clearly visible in the fixed-point structure (see the new section 6). This appears as a rapid (but smooth) change upon crossing the C-H (crossover) line. iv) The C-H mechanism requires vanishing of the (irrelevant) eigenvalue at the C-H line. There is no indication of this close do d=3 (neither in our data nor in the results available in the literature).

We addressed the points from your report in the amended version of the manuscript (the last paragraph of Sec. 4, Sec. 5.3, and Sec. 6).

Sincerely yours, Andrzej Chlebicki, Pawel Jakubczyk

---

## Round 3 · Referee Report · Anonymous (Referee 2) · 2021-3-1

Strengths

Applies FRG approach to a basic question about RG flows in an important model
Explains motivation clearly
Raises interesting questions

Weaknesses

Approximate nature of method prevents definitive answer to the question motivating the study
Data shown is limited to plots of two exponents

Report

This paper applies the functional RG to the O(N) model, aiming to determine the structure of fixed points in the (N, dimension) plane. The motivation comes from work by Cardy and Hamber arguing for a line in this plane where the critical exponents are non-analytic.

Revisiting this prediction using these FRG tools is a worthwhile idea, as the Cardy-Hamber idea has not been sufficiently tested and extended. This paper is a useful contribution. It is well motivated in the introduction, and clearly written (though concise about some technical details). However, the insight provided is limited, as the paper does not give a clear answer to the basic motivating question about the correctness of the Cardy-Hamber prediction. This is mostly because of the approximate nature of the FRG approach (which also involves some ad hoc tuning as implemented) and the difficulty in quantifying its error. The authors state that they do not see the predicted nonanalyticity when the dimension is larger than 2, but acknowledge that it could be that this nonanalyticity is smoothed in the approximate calculation.

In the absence of a definitive conclusion on this issue, the paper gives approximate information about exponents in the (N, dimension) plane and shows that the FRG is capturing quite a lot of the basic physics. Reopening questions about the CH prediction, and about the accuracy of the FRG approach in various limits, may in itself be a valuable contribution.

It is hard for me to judge whether this is sufficient for publication in SciPost. My impression is that this journal is intended to be significantly more selective than a standard Physical Review journal. If so, the answer is probably no. The case might be strengthened if the authors could push their data a little further, for example to provide information about the subleading exponent, which should vanish on the CH line, or to give interesting qualitative information about the shape of the effective potential.

Comments:

- Irrelevant exponent.

The authors search for the CH line by looking for divergent derivatives of exponents. Taking the CH equations at face value would give a step discontinuity in the derivative of 1/nu as the CH line is crossed, with 1/nu and its derivatives being finite on either side of the step. This is to be compared with Fig 7. From this plot it seems hard to rule out the possibility that the locations marked by stars are such nonanalyticities, rounded by errors caused by the approximations used. For example, the magnitude of the errors in Fig 5 is not all that small.

As an alternative way to look for the CH line, is it possible to examine the irrelevant (negative) RG eigenvalue numerically, as done in some FRG works? This should vanish as the CH line is approached.

- Qualitative information about the effective potential? The figures that the authors show focus on exponents. Does the shape of the effective potential show any interesting change as, say, d is varied at fixed N, going from one side of the CH line to the other?

- Sources of error. Is the truncation of the derivative expansion the only source of error, or are there others? Can we assume that the numerical solution of the truncated equations is essentially exact? For readers unfamiliar with the method, it might be useful to have a concise statement of where possible error can come from.

The method is not exact for the power law phase at N=2, d=2, despite the fact that this fixed line has an exact representation using a Lagrangian with two derivatives. Does this indicate that there are additional sources of error, beyond the truncation?

- Another concern is that the method has to be tuned in a somewhat ad hoc way to reproduce the expected behavior at N=2, d=2. If this is necessary at this point, why should the method be expected to accurately reproduce the structure of the flows at other points?

- The authors mention that the equations cannot be solved when N>2 if the dimension is too close to 2. Could the authors comment in the text on what goes wrong?

- Can the authors comment in the manuscript on the level of difficulty in going to the next order in the derivative expansion?

- p11 “These results are compared to the predictions of the epsilon expansion”. Does this mean the direct result from the “traditional” 2+epsilon expansion, or does it mean some kind of resummation taken from the reference cited [38]? This resummation may assume that the exponents are analytic between d=2 and d=4.

- p6: The statement about “exactly controlled limits” being recovered should be clarified. It could give the impression that the method is exact for d=2, N<=2 which as the authors explain is not the case.

  • validity: -
  • significance: -
  • originality: -
  • clarity: -
  • formatting: -
  • grammar: -

Author:  Andrzej Chlebicki  on 2021-04-01  [id 1341]

(in reply to Report 2 on 2021-03-01)
Category:
answer to question
reply to objection
correction

Dear Referee, thank you very much for carefully reviewing our manuscript and preparing the report. We extended the paper in the two directions proposed by you. The obtained results allowed us to strengthen our conclusion and claim that the existence of the nonanalyticity line of C-H is excluded except (perhaps) for the close vicinity of d=2, where our truncation is not sufficiently accurate.

We first briefly comment on the 'weaknesses' listed in your review. - The method implemented by us is indeed approximate, but accurate. When extended to the 4-th order (DE4) (which was achieved only last year in Refs. 21 and 22) it also provides ways of controlling the error bars. Note that in d=3 the DE4 yields estimates of the critical exponents of accuracy comparable to Monte Carlo. In addition, these predictions are compatible with the values from conformal bootstrap. The amended version of the paper gives a definite answer to the question motivating the study except for the close vicinity of d=2. - In the amended version of the manuscript we present results illustrating the evolution of the profiles of the functional fixed points when varying d and N across the crossover as well as a calculation of the irrelevant exponent. We are grateful for the suggestion concerning enriching the paper in these directions.

We now comment on the points from your report: 1. In the present version of the manuscript we provide an analysis of the irrelevant exponent lowering dimensionality starting from d=3. Down to d≈2.2, we find no indication of its vanishing at any value of N. Our truncation is not sufficient to properly resolve the limit d->2 concerning this respect. However, for a broad vicinity of d=3 it gives a fully reliable negative answer as to the possibility of a collision with another fixed point as anticipated by the C-H study. Recovering the limit d->2 is a topic of our ongoing research, which however must go beyond the present approximation level. The corresponding analysis is contained in the new Sec. 5.3. 2. Both the effective potential and the longitudinal stiffness exhibit interesting and rapid (however smooth) changes upon crossing the C-H (crossover) line. The corresponding analysis is contained in the new Sec. 6. 3. For all the practical purposes of the present paper one may treat the truncation of the derivative expansion as the only really relevant source of errors. We verified this by confronting the results obtained by the stability matrix analysis against those obtained from direct integration of the flow. We also extensively tested the dependencies of our results on the grid size, discretization, way of implementing the discrete derivatives, integration accuracy, etc. We comment on this in the new version of the manuscript (p.6, "We have extensively tested..."). 4. The algebraic phase in d=N=2 may be exactly recovered in our framework by freezing the longitudinal mode (see Ref.29). Inclusion of this mode is however necessary to capture the transition. The finite (even though enormously large) correlation length in the low-T phase is due to truncation (not errors of other sources). 5. Upon tuning the regulator in the way implemented by us, the KT singularity is well captured. In particular, this yields the value of \eta, the universal stiffness jump, and the essential singularity of the correlation length in the high-temperature phase. For a non-tuned ('arbitrary') cutoff, our framework yields the KT transition in the form of an extremely sharp crossover (See Refs 28,29). In the low-T phase, the correlation length is 'astronomically' large, and in a realistic experiment or simulation, the two pictures are not distinguishable. This behavior drastically changes when going away from d=N=2 for all the tested regulators. We commented on this in the amended version of the manuscript (see p. 8 "We also point out..."). 6. The numerical problems encountered for d close to 2 and N>2 are related to very large derivatives of the effective potential and the longitudinal stiffness z(\rho) as well as the presence of a pole of the propagator. We commented on this in the new version of the paper (Sec.6, two paragraphs beginning with "The analysis of the ..."). 7. The effort of going to order DE4 is tremendous and this was achieved only last year (Ref. 20). Our motivation for such a study in the present context is in fact quite limited. We are convinced that the picture obtained at order DE2 and DE4 would yield the same reliable conclusions when the close vicinity of d=2 is excluded. Note in particular that the values obtained for the irrelevant exponent (Fig. 10) at order DE2 and DE4 are close from the point of view of our requirements. On the other hand, we do not expect that the limit d->2^+ would be exactly resolved at order DE4 (although improvement may certainly be anticipated). We commented on this in the amended manuscript (see p.7 "An extension of this study to the fourth-order...") 8. We used a direct result from the "traditional" 2+epsilon expansion. We changed the citation, so to cite the original source (Ref 40). 9. We removed the sentence in question.

We would like to thank you once again for your helpful report. We are particularly grateful for your suggestions for amendments, which directed us in a way allowing for strengthening the major conclusion of the paper.

Sincerely yours, Andrzej Chlebicki Pawel Jakubczyk

---

## Round 4 · Referee Report · Slava Rychkov (Referee 1) · 2021-4-3

Report

I thank the authors for carefully revising their manuscript and for adressing my concerns. I now agree with them that their method captures vortices; my previous remark about this was incorrect. I recommend the paper for publication.
  • validity: high
  • significance: high
  • originality: high
  • clarity: high
  • formatting: excellent
  • grammar: excellent

Author:  Andrzej Chlebicki  on 2021-05-11  [id 1415]

(in reply to Report 1 by Slava Rychkov on 2021-04-03)

Dear Professor Rychkov,

thank you very much for reviewing our paper again. We are very pleased about your positive assessment and recommendation.

Sincerely yours,
Andrzej Chlebicki
Pawel Jakubczyk

---

## Round 4 · Referee Report · Anonymous (Referee 2) · 2021-4-9

Report

The authors have made significant additions to the manuscript. These fully address my comments (I thank the authors for the detailed explanations in their reply also). The new results added about the shape of the effective potential and the subleading eigenvalue are interesting and strengthen the results. Fig 10 makes the challenge to the C-H prediction very clear.

I am happy to recommend publication. I have only a few comments for the authors to consider in relation to the new version.

  • On p15 the authors state “whenever the vortex excitations are irrelevant, the relevant eigenvalues of the noncompact CP and Heisenberg universality classes should coincide. In d=3, this should happen for N≳6 according to our results”.

The relationship between the O(N) model and a noncompact CP^m model is special to the case N=3, m=1, so I do not understand how the above statement can be correct.

  • Instead, the relationship between O(3) and noncompact CP1 gives another argument against the claim that the exponents are analytic for all N and d>2.

The standard 2+epsilon expansion of the O(N) model, continued to d=3, can be argued to describe the CP1 model. We know that this theory is distinct from O(3). This implies that at least for N=3, the standard 2+epsilon expansion must fail to describe the O(3) model for epsilon larger than some epsilon, with epsilon smaller than 1. This argument was made in https://journals.aps.org/prx/pdf/10.1103/PhysRevX.5.041048.

This argument is independent of the C-H expansion (though consistent with the C-H picture) so appears to be a challenge to the simplest interpretation of the present numerical results.

  • Regarding topological excitations in d=3, N=3, some early numerical work by Kamal and Murthy, giving hints of novel universal behavior, is https://journals.aps.org/prl/abstract/10.1103/PhysRevLett.71.1911. There were even earlier studies, e.g. https://iopscience.iop.org/article/10.1088/0305-4470/21/1/009/meta, but these used a too-restrictive definition of the model, so did not see a transition in the absence of topological excitations.
  • validity: -
  • significance: -
  • originality: -
  • clarity: -
  • formatting: -
  • grammar: -

Author:  Andrzej Chlebicki  on 2021-05-11  [id 1416]

(in reply to Report 2 on 2021-04-09)
Category:
remark
correction

Dear Referee,

thank you very much for reviewing our paper and preparing the second report. We are very happy about your recommendation for publication. In answer to the points raised in your second report: - The sentence mentioned by you is removed in the new version of the manuscript. - We are grateful for pointing out the paper by Nahum et al, which we find greatly inspiring (also for future investigations). We recognize the substantial weight of the argument concerning the nature of the 2+epsilon expansion and the relation between the CP^1 and O(3) models. Based on the numerical results at hand we would be able to give only very speculative proposals for resolution of this issue and we prefer to leave it as an open problem (as presented in the new version of the manuscript - see the final part of Sec. 5). - We included a reference to the work by Kamal and Murthy.

Sincerely yours, Andrzej Chlebicki Pawel Jakubczyk

---

## Round 4 · Referee Report · Anonymous (Referee 3) · 2021-4-30

Strengths

1) Precise numerical study of second order $O(\partial^2)$ derivative expansion of the Wetterich equation for $O(N)$-models in the $(d,N)$ plane.

2) Accurate analysis (within the approximations of point 1),of the critical exponents $\nu(d,N)$ and $\eta(d,N)$ for continuous values of $d$ and $N$.

3) Study of the sub-leading RG exponent in relation with the Cardy-Hamber line.

Weaknesses

1) The numerical results (i.e. the core of the study) are presented in plots which can be improved in quality and in ease of reading.

Report

The paper deals with the study of $O(N)$ models using the Non-Perturbative Renormalization Group approach based on the Wetterich equation.

The novelty of the research lies in the completeness and quality of the numerical integration of the partial differential equations that encode the RG flow at the $O(\partial^2)$ order of the derivative expansion for continuous values of the parameters $(d,N)$, and in particular around the dimensions of physical relevance $d=2,3$.

The authors investigate the Cardy-Hamber prediction in quite a detail and and claim that their results demonstrate the nonexistence of the Cardy-Hamber line in the vicinity of three dimensions in disagreement with the original work on the topic. This is clearly an important result if fully confirmed.

The work is interesting and novel and definitely deserves publication in this journal, but I first I will ask the authors if they can produce plots of better quality (more clear, more readable, better scaled and with a color-code that will function also in print). Since all the outputs of this type of study are numerical and are displayed graphically, improving this point will make the manuscript much more understandable to readers and of greater general value.

A couple of more specific questions are: 1) why is the comparison between the exact and the computed $\eta$ in Figure 5 off by an almost a constant factor? Furthermore I suggest to the authors to graph the whole range $-2\leq N \leq 2$ since the point $(0,-2)$ is was solved by Fisher long ago. 2) Can the authors display the equations they are actually solving or they are really so complicated?

Requested changes

1) Improve of the plots as discussed in the report.

  • validity: top
  • significance: top
  • originality: high
  • clarity: high
  • formatting: good
  • grammar: good

Author:  Andrzej Chlebicki  on 2021-05-11  [id 1417]

(in reply to Report 3 on 2021-04-30)
Category:
remark
answer to question

Dear Referee,

we would like to thank you for reviewing our paper. We are very glad about your positive recommendation.

We introduced modifications to the plots. We hope it somewhat improved their readability. Concerning the specific points from the report: - We, unfortunately, do not have an explanation why the two curves plotted in Fig.5 are shifted by roughly a constant number over a range of values of N. However, please observe that this no longer holds true for N approaching 2, which is the region of our major interest. Our study is restricted to positive values of N (as explained and illustrated in particular in Fig.1). For consistency, we, therefore, refrain from discussing N negative also in Fig. 5. We agree that N<0 (and also the range d\in (1,2]) is also interesting and would like to address it in the future. - The flow equations at the considered truncation order fit roughly one page. In the amended version of the manuscript, we added an appendix, where they are exposed.

We thank you once again for reviewing our work.

Sincerely yours, Andrzej Chlebicki Pawel Jakubczyk

---

## Round 4 · Author Response

Dear Editor,

thank you very much for making the reports available to us. We amended
the paper following the Referees' suggestions. Below we summarize the
introduced extensions and alternations and give our response to the
Referees' reports.

Sincerely yours,
Andrzej Chlebicki
Pawel Jakubczyk

---

## Round 4 · List of Changes

We significantly strengthened our conclusion, which is reflected in the new Abstract as Summary, as well as changes throughout the text. - We modified and extended the introduction and summary, restructured Sec. 2, introduced minor changes to Sec. 3, added the last paragraph of Sec. 4, slightly modified the end of Sec. 5.1. - We added the new Sec. 5.3 and Sec. 6.

---

## Round 5 · Author Response

Dear Editor,

thank you very much for making the reports for our paper available. In the presently resubmitted version, we addressed most of the comments/suggestions from the Referees.

We attach the summary of changes and provide a response for all three of the Referees.

We are grateful to you for handling our paper.

Sincerely yours,
Andrzej Chlebicki
Pawel Jakubczyk

---

## Round 5 · List of Changes

Summary of changes: - We made minor modifications (mostly stylistic) throughout the text. - We replaced most of the figures for better readability. - We significantly modified the final part of Sec.5. - We replaced Fig. 7. The previous version had been (by mistake) plotted with the epsilon expansion results from the previously cited article Kleinert 1998, which (unexpectedly) differs from the currently cited Bernreuther 1986 in epsilon^4 coefficients. - We replaced Fig.10. The previous version had been (by mistake) plotted with a less accurate set of data points. The new points are slightly shifted down, agree even better with DE(4)/MC results, and are situated even further from e_2 = 0. - We added the appendix. - We added references 6, 41, and 44.

---

## Editorial Decision

published